# CD137 as an Attractive T Cell Co-Stimulatory Target in the TNFRSF for Immuno-Oncology Drug Development

**DOI:** 10.3390/cancers13102288

**Published:** 2021-05-11

**Authors:** Kenji Hashimoto

**Affiliations:** Crescendo Biologics, Ltd., Meditrina Building 260, Babraham Research Campus, Cambridge CB22 3AT, UK; khashimoto@crescendobiologics.com

**Keywords:** CD137, T cells, bispecific, clinical trial

## Abstract

**Simple Summary:**

CD137 is an interesting immuno-oncology target. The recent advances in CD137 targeting technologies to mitigate toxicity while maintaining potency have made this space even more attractive. In this article, our current understanding of CD137 biology is reviewed along with data from clinical trials targeting T cell co-stimulatory receptors in the TNFRSF. Next generation CD137 targeting molecules currently in early clinical development are also reviewed. Finally, the future challenges of CD137 targeting molecules are discussed.

**Abstract:**

Immune checkpoint inhibitors have altered the treatment landscape significantly in several cancers, yet not enough for many cancer patients. T cell costimulatory receptors have been pursued as targets for the next generation of cancer immunotherapies, however, sufficient clinical efficacy has not yet been achieved. CD137 (TNFRSF9, 4-1BB) provides co-stimulatory signals and activates cytotoxic effects of CD8^+^ T cells and helps to form memory T cells. In addition, CD137 signalling can activate NK cells and dendritic cells which further supports cytotoxic T cell activation. An agonistic monoclonal antibody to CD137, urelumab, provided promising clinical efficacy signals but the responses were achieved above the maximum tolerated dose. Utomilumab is another CD137 monoclonal antibody to CD137 but is not as potent as urelumab. Recent advances in antibody engineering technologies have enabled mitigation of the hepato-toxicity that hampered clinical application of urelumab and have enabled to maintain similar potency to urelumab. Next generation CD137 targeting molecules currently in clinical trials support T cell and NK cell expansion in patient samples. CD137 targeting molecules in combination with checkpoint inhibitors or ADCC-enhancing monoclonal antibodies have been sought to improve both clinical safety and efficacy. Further investigation on patient samples will be required to provide insights to understand compensating pathways for future combination strategies involving CD137 targeting agents to optimize and maintain the T cell activation status in tumors.

## 1. Introduction

CD137, or 4-1BB/tumor necrosis factor receptor superfamily (TNFRSF9) was first reported in 1989 as an inducible gene in T cells through cDNA sequencing analysis [1]. Later, it was found that CD3 stimulation could induce CD137 expression where antibody-mediated cross-linking of CD137 resulted in T cell proliferation [2]. CD137 is expressed on both CD4^+^ and CD8^+^ T cells within 24 h following stimulation, peaking at around 2–3 days [3]. Human CD137 mRNA is induced in activated T and B cells as well as monocytes and other cell lineages [4]. It is now accepted that CD137 expression is also seen on natural killer (NK) cells and dendritic cells (DCs) as well as other immune cells [5].

CD137 is one of the TNF receptor family (TNFRSF) members. The gene for CD137 is located in the chromosome 1p36 locus which also encodes other co-stimulatory TNFRSF members [6,7]. TNFRSF members are characterized by the ability to bind TNFs via a disulfide bonds forming cysteine-rich extracellular domain [8]. Like other TNFRSF members, three monomeric CD137s bind to a trimeric CD137L to activate intracellular signaling [9]. There are two distinctive classes of TNFRSF function: death receptors (DR) that ultimately cause caspase activation and cell death such as Fas receptor, DR4 and DR5; and TNF receptor associated factors (TRAFs) that activate NF-κB leading to cell survival such as CD137, glucocorticoid-induced tumor necrosis factor (GITR) and OX40 [7].

CD137 is expressed on CD4^+^ and CD8^+^ T cells rapidly after exposure to antigen and it has been shown that cross-linking of CD137, and the T cell receptor (TCR) on activated T cells can deliver co-stimulatory signals to T cells resulting in T cell proliferation, survival, memory formation and stronger effector function for cytotoxicity and cytokine production [10] (Figure 1). The effect of T cell proliferation seems more pronounced in CD8^+^ T cells compared with CD4^+^ T cells. By contrast, CD28, the constitutively expressed co-stimulatory receptor on T cells activates CD4^+^ T cells more than CD8^+^ T cells [11]. This preferential proliferation of CD8^+^ T cells makes CD137 a particularly attractive therapeutic target for anti-cancer treatment. Co-stimulatory receptors have minimal signaling by themselves but are essential to drive T cells to respond to antigens presented on tumor cells. In fact, the absence of co-stimulation leads to clonal deletion that protects the host against unwanted immune responses. In addition, suboptimal co-stimulation could result in a state of non-responsiveness or anergy [12]. The clinical success of chimeric antigen receptors (CARs) encompasses both the signal 1 element of CD3-TCR interactions and the signal 2 element of co-stimulation, and CD137 is so far reported to have the best effectiveness among co-stimulators in construction of CARs against leukemia [13].

## 2. The Role of CD137 in the Tumor Microenvironment

### 2.1. Prevalence of CD137-Positive T Cells in Tumor Microenvironment

Tumor-infiltrating lymphocytes (TILs) are heterogeneous lymphocyte population in tumors that target a variety of antigens. Interestingly, CD8^+^ TILs have enhanced expression of PD-1, TIM3, CD137 and LAG3 (% of each expression is 32.4%, 13%, 10.8%, 5.7%, respectively) in melanoma patients and nearly 80% of CD8^+^CD137^+^TILs express PD-1 [21]. PD-1^+^CD137^+^ TILs represent approximately 10% of TILs and both PD-1^+^CD137^+^ and PD-1^+^CD137^−^ TILs can recognize the tumor while PD-1^−^CD137^−^ TILs cannot [21]. However, another report suggests that only CD8^+^CD137^+^ TILs, regardless of PD-1 expression can recognize tumor in ovarian cancer or melanoma [22]. Given the populations of PD-1^+^ TILs are high, it is reasonable to target PD-1^+^ TILs in cancer immunotherapy. However, these PD-1^+^ TILs may be exhausted. In HCC, PD-1^+^ TILs almost exclusively express CD137 and CD137 expression in CD8^+^ TILs is the highest numerically across other co-stimulatory receptors. Importantly, CD137 agonistic antibodies showed an enhancement of CD8^+^ TILs in function as well as the anti PD-1–mediated reinvigoration of CD8^+^ TILs. The report also examined CD137^+^ TILs in 6 different cancers and HCC had the highest number of CD137^+^ TILs in the examined tumors (ovarian cancer, NSCLC, intrahepatic cholangiocarcinoma, colorectal cancer and glioblastoma) [23].

### 2.2. The Role of CD137 in the Context of Clincial Trials

Costimulatory receptors in the TNFRSF including CD137, OX40, CD27 and GITR can promote proliferation and survival in T cells, increase effector activity including cytokine production and drive the generation of T cell memory. The question remains unanswered whether all of these costimulatory receptors need to be involved for T cell activation for anti-tumor response or whether T cell response is primarily driven by one or two receptors and others are complementary [24]. It is possible that CD8^+^ T cell activation compared to CD4^+^ T cell activation may be more important for immediate cytotoxic effects when targeted with CD137 or CD27 agonists. For sustainable anti-tumor response, generation of memory T cells is also important and CD137, CD30 or OX40 stimulation are known to provide survival signals or induce memory on T cells, while a deficiency of CD137 ligand (CD137L) or OX40 ligand (OX40L) results in a decrease of CD8^+^ memory T cells in bone marrow [25,26,27]. CD137, OX40, CD27 and HVEM also appear to contribute effector memory T cell (Tem) and central memory T cell (Tcm) expansion [28,29,30]. CD137 as well as OX40, TNFR2 (TNFRSF1B) and GITR have been shown to block naïve T cells from differentiating into FOXP3^+^ T regulatory cells (Treg) but it has also been reported that stimulation of these receptors may induce Treg [7]. The best way to address the question as to which co-stimulatory receptors are critical in tumor killing, clinical trials of systemic agonistic antibodies targeting these costimulatory receptors in cancer patients, will provide further insights. In the following section, each costimulatory therapy target will be discussed (Table 1).

#### 2.2.1. CD27

Varlilumab is an agonistic antibody against CD27 that showed 1 complete response (CR) in Hodgkin lymphoma and the response remained >33 months even after discontinuation of the treatment, while no responders were observed in other hematological malignancies [31]. Varlilumab is currently being investigated in combination with other agents including rituximab, nivolumab in B cell lymphoma or atezolizumab with radiation in non-small cell lung cancer [49]. MK5890 is another CD27 agonistic antibody currently being investigated in phase I studies either as monotherapy or in combination with pembrolizumab in patients with advanced solid tumors. Partial responses (PRs) and CRs have been reported and further investigation is warranted as both agents appear to be tolerable. Overall, limited data is available for CD27 targeting molecules but monotherapy does not appear to induce robust clinical responses so far although the therapy is tolerated.

#### 2.2.2. OX40

OX40 has been shown to promote CD4^+^ and CD8^+^ T cell survival and expansion as well as controlling effector memory T cells. 9B12 is a murine agonistic anti-human OX40 monoclonal antibody (mAb) assessed in patients with advanced solid tumors as a phase I study. While some level of tumor regression was observed, no confirmed responses were reported. An increase of Ki67 was seen in peripheral CD4^+^ T-cells on day 8 and CD8^+^ T cells on day 15 and there was no impact on CD4^+^FoxP3^+^ Treg cells. Immunogenicity to this murine antibody in human patients precluded multi-cycle therapeutic dosing. It is yet unclear whether there was an increase of T cells specific to tumor and thus, tumor tissue sample analysis is warranted [50].

PF-04518600 is a humanised IgG2 agonistic mAb against OX40. In a phase I study, only 1 patient responded to PF-04518600 out of 25 patients [51]. PF-04518600 was further investigated in patients with hepatocellular carcinoma but no RECIST responses were reported [33].

MEDI0562 is an agonistic humanized IgG1 mAb against OX40 investigated in advanced solid tumors which reported 2 immune partial responses out of 55 patients [34]. MEDI0562 in combination with durvalumab or tremelimumab was assessed for safety and efficacy in advanced solid tumors. While 3 responders in combination with durvalumab (n = 26) were observed, 2 patients died due to renal failure and more than 70% of patients developed anti-drug antibodies. There was an increase of CD8^+^ memory T cells observed [52]. MEDI6469 is a murine IgG mAb against OX40 was tested in phase I study but no responder was observed [50]. MEDI6469 was further investigated in HNSCC as a neoadjuvant therapy. Paired tumor biopsy samples showed that an increase in the tumor-reactive CD103^+^CD39^+^CD8^+^ TILs could serve as a potential biomarker of anti-OX40 clinical activity [35]. Interestingly, a highly potent engineered human OX40 ligand IgG4P Fc fusion protein was investigated preclinically and in cynomolgus macaques [53], but has not yet entered human clinical studies to date.

PF-04518600 in combination with utomilumab in patients with advanced melanoma or NSCLC who progressed after checkpoint inhibitor treatment was assessed. One NSCLC patient achieved partial response lasting at least 6 months, while 70% of melanoma patients and 35% of NSCLC patients achieved stable disease for a median duration of 16 weeks and 24.1 weeks respectively. Paired tumor biopsy analysis showed greatest CD8^+^ T cells increase in NSCLC patients who achieved the longest stable disease durations [54].

GSK3174998 is a humanized IgG1 agonistic antibody against OX40 and was assessed as monotherapy or in combination with pembrolizumab in a phase I study. One PR was observed (n = 45) in the monotherapy arm and there were 9 responders including 2 CR in combination arm (n = 96). Tumor biopsy data suggests that increased NK and decreased Treg cells were demonstrated in some responders [36]. Currently, GSK3174998 is being investigated with an anti-inducible T cell co-stimulator (ICOS) agent or with an anti-BCMA agent in multiple myeloma.

MOXR0916 is a humanized IgG1 mAb targeting OX40, assessed as a monotherapy or in combination with atezolizumab and 2 out of 51 patients responded to this combination; nevertheless, the programme has been discontinued [37]. INCAGN01949 is another fully human IgG1 mAb and a deep response to sequential therapy of ipilimumab and nivolumab after INCAGN01949 was observed in 2 cases [38].

BMS-986178 is another fully human IgG mAb against OX40 assessed as monotherapy or in combination with nivolumab in advanced solid tumors. No anti-tumor activity was observed with monotherapy and overall response rate (ORR) was 6–12% in the combination cohort [39]. Overall, OX40 targeting monotherapy appears to be tolerable however, it does not induce durable anti-tumor responses.

#### 2.2.3. GITR

GITR has been an attractive target for immunotherapy given its role in promoting T effector cell (Teff) functions and hampering Treg suppression. Anti-agonistic IgG1 GITR antibody TRX518 showed a reduction of Treg and increase the Teff/Treg ratio. While TRX518 appears to be safe and well tolerated, no substantial responses were seen [40]. Similarly, MK-4166, MK-1248, BMS986156, MEDI1837 and AMG288 were well tolerated but effective therapeutic indications are yet to be identified due to low clinical response rates to monotherapy [41,42,43,44]. MK-4166, an IgG1 mAb in combination with pembrolizumab was also assessed and 8 out of 13 immune-checkpoint inhibitor naïve melanoma patients achieved a response (ORR, 62%) but no responses were observed in patients that had been previously treated checkpoint inhibitors [41]. MK-1248, an IgG4 mAb in combination with pembrolizumab achieved 3 responses in 17 patients (ORR = 17.6%). BMS-986,156 was further tested in combination with nivolumab but again, no significant response was seen (0–11%) [43]. In summary, targeting GITR as a monotherapy does not look promising and therefore, optimisation of therapeutic combinations or other approaches for activating T cells through GITR is warranted.

#### 2.2.4. CD137

CD137 also provides co-stimulation that leads to T cell activation, proliferation and survival as well as generation of Teff memory, and clinical results with CD137 agonists appear to be different to those from clinical studies interrogating other co-stimulatory targets. Urelumab (BMS-663513) is an IgG4 mAb targeting CD137 and provided clinical responses in B cell malignancies as monotherapy (ORR; 6% in diffuse large B cell lymphoma, 12% in follicular lymphoma, 17% in other B cells lymphomas, respectively) or combination with rituximab (ORR, 35%). Urelumab was also tested as monotherapy in solid tumors and responses were observed in melanoma patients [55]. However, many of the responses were seen above the maximum tolerated dose (MTD) level and there was a dose-dependent liver toxicity observed [46,47]. Hepatotoxicity appears to be associated with the agonistic ability of urelumab to activate liver Kupffer cells [56]. Therefore, clinical response has been sought with lower doses of urelumab that do not cause significant hepatotoxicity in combination with other therapeutic agents. Urelumab when combined with nivolumab and the cancer cell vaccine GVAX in a neoadjuvant/adjuvant context achieved a 30% pathological response rate in resectable pancreatic cancer without causing significant safety issues. Interestingly, these responses were not observed in patients treated with GVAX alone or GVAX plus nivolumab [57], suggesting a significant role for T cell activation mediated by targeting CD137 in clinical response.

Utomilumab (PF-05082566) is an IgG2 mAb CD137 agonist with a better safety profile than urelumab as its potency against CD137 is weaker. However, due to the reduced potency, ORR with monotherapy was 3.8% for solid tumors and 13.3% for Merkel cell carcinoma [48]. Soluble forms of CD137 (sCD137) were measured in this study and an increase of sCD137 was observed following treatment with utomilumab. The peak of sCD137 was within 50–100 h of utomilumab administration and there was no dose relationship observed [48]. Since sCD137 acts as an antagonist to the co-stimulatory activities of membrane bound CD137 and reduces immune activity [58], further clarification is needed around how sCD137 impacts the efficacy of CD137 agonists and whether monitoring of sCD137 is helpful in patients who receive CD137 agonists. Utomilumab in combination with mogamulizumab, a humanized mAb targeting C-C chemokine receptor 4 (CCR4) was assessed for safety and efficacy in solid tumors. While the regimen was tolerable, the ORR was only 4.2% [59]. Utomilumab in combination with pembrolizumab was also investigated in solid tumors and the combination demonstrated responses in 6 out of 23 patients including 2 CRs [60]. Utomilumab was combined with PF-8600 (a humanized agonist IgG2 to OX40) but no significant response was observed [54]. Currently, utomilumab is under investigation in combination with CAR-T cells (NCT03704298). Therefore, alternative approaches to reduce hepatotoxicity while maintaining CD137 agonistic potency have been sought.

### 2.3. The Role of CD137 in Other Immune Cells

While CD137 is well investigated in T cells, additional knowledge has been acquired for non-T cells. CD137 is known to be expressed in NK cells [5]. CD137 in NK cells can be induced by IL-2 or IL-15, and CD137 stimulation induces IFN-γ and TNF-α generation. Agonistic mAb to CD137 can promote NK cells to interact with CD8^+^ T cells and support interaction with dendritic cells (DCs) to promote their functions (Figure 1) [14]. However, CD137 functionality on NK cells did not lead to direct tumor killing effects; the tumor killing is performed by CD8^+^CD137^+^ T cells because the antitumor efficacy of a CD137-bispecific molecule in mouse studies was completely diminished in the absence of CD8^+^ T cells, while CD4^+^ T and NK cells were dispensable [61].

CD137 expression on DCs and monocytes is also functional. Upon infusion of agonistic CD137 mAb to CD137^+^ DCs, DCs can enhance T cell-proliferative responses to both alloantigens and viral antigens in mice [62]. DCs produce IL-6 and IL-12 upon stimulation of CD137. Interestingly, monocytes expressing CD137 differentiate to DCs upon CD137L signaling and help maturation of DC cells. Monocytes also induce TNF-α and IL-8 while suppressing IL-10 and potentially induce macrophage M2 polarization as well as leading to B cell apoptosis [5,19,20,63,64]. CD137 is also found on B cells and CD137 stimulation by CD137L or agonistic mAb to CD137 on B cells leads to B cell proliferation and survival, but does not promote Ig class switching [5,20].

The main tumor killing functions dependent upon CD137 stimulation arise from CD8^+^ T cells. Ligands or agonistic mAb to CD137 lead to CD8^+^ T cell proliferation, survival and enhanced cytotoxicity. CD137 stimulation promotes generalized formation of CD8^+^ T memory cells [15]. Along with CD8^+^ T cell stimulation and production of IL-2 and IFN-γ, CD137 stimulation helps CD4^+^ T cells to support CD8^+^ T cell activity including production of IL-2, possibly further enhancing expression of CD137 on NK cells (Figure 1). The role of CD137 in Treg cells still needs more investigation particularly in the human system. A few reports suggest that CD137 mAb agonists appear to increase naïve Treg but also converts the function of Treg to cytotoxic or helper T cell aspects and induce a production of IFN-γ (Figure 1) [16,17]. Moreover, CD137L stimulation inhibits antigen- and TGF-β-driven conversion of naïve CD4^+^FoxP3^−^ T cells into induced Treg cells via stimulation of IFN-γ production by CD4^+^FoxP3^−^ T cells [18]. However, high frequency of CD137^+^ Treg is associated with poor prognosis in lung adenocarcinoma [65] and another report suggests that agnostic antibody to CD137 in gastric cancer has little effect on Tregs [66]. Given that CD137^+^ Tregs are more immunosuppressive than CD137^−^ Tregs in the non-tumor environment [67,68] comprehensive understanding of the role of CD137^+^ Treg in the tumor microenvironment is warranted.

## 3. Technological Advances to Mitigate Hepatotoxicity for CD137 Targeting Molecules

### 3.1. Bispecific Molecules Targeting CD137

Among the clinical trials targeting TNFRSF co-stimulatory receptors, CD137 appears to be the most robust target. To reduce the hepatotoxicity observed with systemic CD137 agonism with urelumab while maintaining potency, restricting CD137^+^ T cell agonism to the tumor microenvironment by targeting CD137 with bispecific format molecules that simultaneously engage tumor associated antigens (TAA) appear to be an ideal approach. First, only antigen experienced T cells (CD137^+^ T cells) are conditionally where there is a target TAA. Therefore, on-target off tumor toxicity may be minimized as would be cytokine release syndrome (CRS) given that only selected T cells are engaged. Second, CD137 targeting bispecific compounds may be more robust to antigen loss than CD3-engaging bispecifics, since CD137 stimulation expands tumor reactive memory T cells independently of MHC or antigen [69,70]. As a result, the memory T cell pool is expanded, and may recognize multiple tumor antigens. Antigen-loss during the course of treatment has been identified as a cause of resistance with CD3-targeting bispecific compounds [71]. Lastly, T cell exhaustion by engaging CD3^+^ T cells may be mitigated by CD137 agonism. The data suggests that CD137 co-stimulation restores the functionality of exhausted CD8^+^ TIL in mouse, prolongs persistence of cytotoxic T lymphocyte (CTL) and enhances CTL function [72,73]. Therefore, durable efficacy can be expected.

Table 2 summarizes current clinical development of CD137-targeting bispecific molecules. PRS-343 is a bispecific molecule targeting HER2 and CD137 with technology using anticalin binding protein fused to whole IgG, currently being investigated in a phase I study for HER2-positive solid tumors. In patients who were treated at active dose levels (n = 33), 12% of patients achieved ORR including 1 CR and the disease control rate (DCR) was 52%. There was no DLT reported. Strikingly, there was also no CRS nor toxicities in central nervous system reported which highlighted a further advantage of CD137-targeting versus CD3-targeting bispecific compounds [74]. PRS-343 in clinical combination with atezolizumab was also reported but the combination appears to be lack of additive effects [75]. Biomarker data of PRS-343 showed an increase of CD8^+^ T cells in tumor as well as sCD137 in serum in a dose dependent manner. An increase of NK cell density was also observed in the biologically effective dose range and clinical responses were observed in low CD8^+^ TIL and low PD-L1 tumors [76].

GEN1046 is a bispecific antibody to PD-L1 and CD137 being investigated in solid tumors. PD-L1 is used as a TAA and GEN1046 also inhibits signaling through the PD-L1 axis. While an MTD was not reached, Grade 3 hepato-toxicities were reported. Interestingly, reduced peripheral immune modulation including Ki67 changes in CD8^+^ T cells were observed at higher dose levels and clinical responses were observed at 80mg, 100mg and 200 mg while up to 1200 mg were tested (4 PRs in 56 patients, DCR 65.6%). This phenomenon where maximal efficacy with bispecific molecules is observed at intermediate dose levels could be explained by a “bell-shaped” pharmacology effect [77]. In cohort expansion, preliminary anti-tumor efficacy in patients with NSCLC previously treated with checkpoint inhibitors was observed [78]. These two studies support the use of CD137 targeting bispecific format molecules in solid tumors.

RO7122290 and RO7227166 are unique trimeric CD137L and Fab fusions to fibroblast activation protein (FAP) or CD19 respectively [79]. The safety and clinical efficacy of RO7122290 alone or in combination with atezolizumab was assessed in a phase I study. Clinical response as monotherapy was limited and likewise did not appear promising when combined with atezolizumab. It is uncertain whether anti-drug antibodies (ADAs) may have negatively impacted efficacy because ADA in 20% of the evaluable patients with loss of efficacy in 40% of the positive patients was reported [80]. RO7227166 is also being investigated in a phase I study.

ES101/INBRX-105 is another PD-L1 and CD137 bispecific compound currently in a cohort expansion part of a phase I clinical study. Clinical efficacy of ES101/INBRX-105 monotherapy has been reported with 8 patients achieving stable disease out of 18 patients treated with a greatest reduction in tumor volume observed to be 20% according to a press release (https://inhibrx.investorroom.com/sec-filings, accessed on 10 May 2021). Combination studies with pembrolizumab are now ongoing. There are additional CD137 targeting bispecific molecules currently investigated in phase I studies. One of these, CB307, uniquely targets prostate specific membrane antigen (PSMA) with a heavy chain only antibody fragment component. FS120 and GEN1042 are also interesting molecules given both simultaneously target two co-stimulatory receptors: OX40/CD137 and CD40/CD137 respectively.

### 3.2. Agonistic Monoclonal Antibodies to CD137 with an Extra Safety Switch

Advanced technologies are used to mitigate activation of Kupffer cells induced by urelumab for some new agonistic mAb to CD137. STA551 is a switch Ig antibody only activated where immunosuppressive extracellular adenosine triphosphate (ATP) presents and is currently being investigated in solid tumors in a phase I study either as monotherapy or in combination with atezolizumab [81].

AGEN2373 is an IgG1 mAb targeting the cystine-rich domain IV of CD137. CD137 signaling is only induced by AGEN2373 upon FcγR cross-linking to an APC and native CD137L/CD137 interactions are still allowed. Therefore, AGEN2373 provides CD137 conditional activation. AGEN2373 is currently investigated in a phase I study as a monotherapy or in combination with balstilimab (anti-PD-1) [82]. LVGN6051 has a weak agonistic action to CD137 but shows similar agonistic activity to urelumab when FcγRIIB-expressing cells are present. Preclinical mouse model studies did not show liver toxicity and LVGN6051 is currently being investigated in a phase I clinical study as monotherapy and in combination with pembrolizumab for solid tumors [83].

## 4. Potential Combination Partners for CD137 Targeting Molecules

### 4.1. Checkpoint Inhibitors

CD137 targeting bispecific molecules are attractive agents with less safety concerns for CRS than CD3 targeting bispecific molecules and may be able to provide prolonged efficacy of T cell activation even a TAA is lost. Previous clinical studies of HPN424 and RO7122290 showed the potential of combination studies with PD-L1 inhibitors [75,80]. Utomilumab in combination with pembrolizumab was assessed in 33 patients with solid tumors and the combination achieved a 26% ORR including 1 CR, however, the effect of the combination compared with pembrolizumab monotherapy is unclear [60]. Urelumab in combination with nivolumab was also assessed in melanoma and other malignancies. The combination achieved 50% of best response rate in melanoma regardless of PD-L1 status while the response was not remarkable in other malignancies. The urelumab dose used for this study was 8mg/kg, which was above MTD level [84]. CD137 bispecific molecules with similar potency to urelumab in combination with PD-1 blockade should be assessed in clinical trials because CD137^+^ TILs also express PD-1, and CD137 bispecific molecules in combination with anti-PD-1 mAb demonstrated the best synergistic effect in preclinical model systems, when compared with either CTLA-4 or TIM3 targeting molecules by enhancing terminally differentiated CD8^+^ T cells in tumor [61]. In addition, T cell activation via CD137 is known to increase metabolic capacity in tumor microenvironment and CD137 pretreatment is sufficient to provide metabolic support for a full anti–PD-1 response [85].

### 4.2. CD3 Targeting Bispecific Compounds

Another interesting CD137 combination possibility is with CD3-targeting bispecific molecules. Given that CD3 stimulation can provide signal 1, co-stimulation involving a CD137 targeting bispecific molecule could provide durable anti-tumor response. Importantly, the CD3 and CD137 bispecific molecules do not have to target the same TAA. In a preclinical mouse model, the combination of bispecific molecules targeting CD3/CEA and CD137L/FAP showed synergistic anti-tumor effects and this was accompanied by an increase of more CD8^+^ T cells in tumor than with a CD3/CEA bispecific molecule only [79]. Recently, a phase Ib study of RO7122290 in combination with cibisatamab (CEA x CD3 bispecific molecule) and obinutuzumab has been initiated in colorectal cancer (NCT04826003). Sequence and timing of CD3 and CD137 targeting molecules will need to be optimized in order to maximize both efficacy and safety in a clinical setting.

### 4.3. Anti-Angiogenic Compounds

Pathologic angiogenesis is a key feature of tumor biology and shares similar mechanisms to the physiological angiogenesis processes. However, tumor-mediated angiogenesis is characterized by a failure of the resolution phase, which leads to the generation of a highly disorganized vascular network [86]. Angiogenesis is critical for persistent tumor growth and metastasis and dysregulation of angiogenesis creates high interstitial fluid pressure (IFP) in the tumor, which results in hypoxia, low pH which inhibits immune cell activation, poor blood supply, and impairs the delivery of systemic administered therapies to tumors [87]. As a result, the delivery of systemic anti-cancer treatment becomes heterogeneous in tumor due to these physiological barriers. Indeed, vascular re-normalization in tumor via vascular endothelial growth factor (VEGF) targeting inhibitors can restore the balance of antiangiogenic and proangiogenic factors and revert the vasculature to a more normal phenotype. In fact, IFP was decreased by 70% and vascular density was decreased by 50% for patients treated with bevacizumab [88]. Since immunotherapy in cancer treatment relies on the accumulation and activity of immune cells in the tumor microenvironment and immune response and vascular normalization seem to reciprocally regulated, it is reasonable to combine immunotherapy and anti-angiogenic agents. For example, protumoral M2-like macrophages, T helper 2 (T_H_2) cells and Treg cells secrete pro-angiogenic factors and promote vascular immaturity, while CD8^+^ T cells and CD4^+^ T_H_1 cells suppress angiogenesis and induce vascular maturation by secreting IFN-γ. However, CD8^+^ T cells can have difficulty infiltrating the tumor microenvironment due to malformed tumor vessels [89]. Anti-angiogenic molecules in combination with checkpoint inhibitors have been documented to provide enhanced anti-tumor activities in several cancers [90,91,92,93]. Similarly, anti-tumor effects induced by CD137 targeting molecules may be enhanced by combinations with anti-angiogenic agents.

### 4.4. ADCC Enhanced Antibodies

Targeting CD137 may also enhance the efficacy of antibody-dependent cell-mediated cytotoxicity (ADCC) dependent mAb treatment because of the activation mechanism of the NK cellsUrelumab and rituximab combination treatment was assessed in patients with B cell lymphoma. Responses were observed although many of them were at drug exposure levels above the MTD [46]. Gopal et al. reported that the ORR was 21.2% in non-Hodgkin lymphoma patients treated with utomilumab in combination with rituximab [94]. Further investigation is warranted to see whether CD137 agonistic compounds enhance ADCC activity in humans to confirm this pre-clinical hypothesis. To this end, PRS-343 will be investigated in combination with ramucirumab and paclitaxel in HER2-positive gastric cancer.

## 5. Discussion

### Challenges of CD137 Targeting Molecules in Solid Tumors

Several challenges in cancer immunotherapy are highlighted in the article written by Hedge and Chen [95]. In particular, there is a need to establish a translatable preclinical model that can predict clinical safety and response. It is also important to understand organ specific immune response in addition to immune response by cancer type. In addition, there are aspects around optimizing immune cell activation and involving optimization of the molecular size of therapeutic agents to enable tumor penetration and activate T cells in the tumor milieu which are critical. For penetration of the therapeutic agent into tumor, smaller is, in the general case, better [96,97]. However, if the size of the molecule is small enough to be cleared by the kidney, the therapeutic molecule may not persist sufficiently long in the circulating system to reach tumors effectively [88]. To enable better drug delivery by reducing renal clearance for smaller bispecific biologic compounds, a compound binding to albumin can be employed. Albumin, the most abundant and long-lived serum protein, exhibits desirable features as a payload carrier for therapeutic biologics. Albumin has been shown to accumulate within the tumor environment or inflamed tissues by receptor-mediated active transport as well as passive transport. Albumin has been used successfully as a half-life extender [96,97]. CB307 and NM21-1480 use human serum albumin (HSA) in their CD137 targeting bispecific format and are currently being investigated in phase I clinical trials (Table 2). However, determination of which therapeutic half-life gives the best opportunity to engage T cells remains unclear since continuous exposure of T cells to antigen may lead T cell exhaustion. Clinical investigation of multiple dosing schedules may be required to answer this question.

Optimization of patient selection for CD137 targeting molecules in clinical trials also needs to be explored. CD137^+^ TILs in a variety of solid tumors have been explored by mRNA level and the report showed that the tumor types having high mRNA of CD137 appear to be similar to the tumors with high TIL scores [98,99]. To date, no clinical trial of CD137 targeting agents have been conducted prospectively to enrich patient populations based on CD137^+^ TIL information. CD137 is expressed on antigen experienced T cells but the dynamism of CD137 expression makes a definitive determination difficult to capture, therefore, it may not be feasible to utilize this information effectively for patient selection. Indeed, there is no data reporting response of mAb to CD137 in association with CD137 expression level in the tumor microenvironment while some responses were seen in the tumor with low CD8^+^ T cells for patients treated with PRS343 [76]. Instead, there will need to be alternative methods to identify tumors in which T cells acquire signal 1 through MHC-TCR interactions on an ongoing basis. For PRS-343, the clinical responses were seen at the dose level where CD8^+^ T cell expansion was observed, however, CD8^+^ T cell expansion may prove to be a necessary, but not sufficient condition for clinical efficacy, as clinical responses were not seen in other patients even though CD8^+^ T cell expansion was observed [76]. Understanding compensating pathway to maximize the T cell cytotoxic activity while maintaining a suitable safety margin will need further investigation to pave the way for successful combination therapy. Finally, having productive way of generating memory T cells, for example though CD137 agonism, could be a key for long lasting anti-cancer immune efficacy as memory T cells do not get exhausted.

## 6. Conclusions

CD137 as an immune-oncology target remains attractive and still one of the most promising targets among T cell co-stimulatory receptors in the TNFRSF based on the available clinical data. Current phase I studies of next generation CD137 targeting molecules will address the issue of hepato-toxicity without compromising potency. Understanding biology upon CD137 treatment in patients and the strategy for patient selection for optimizing clinical response are critical for the future success of CD137 targeting agents. CD137 target molecules also support combination with other anti-cancer agents to enhance immune cell responses.

## Figures and Tables

**Figure 1 cancers-13-02288-f001:**
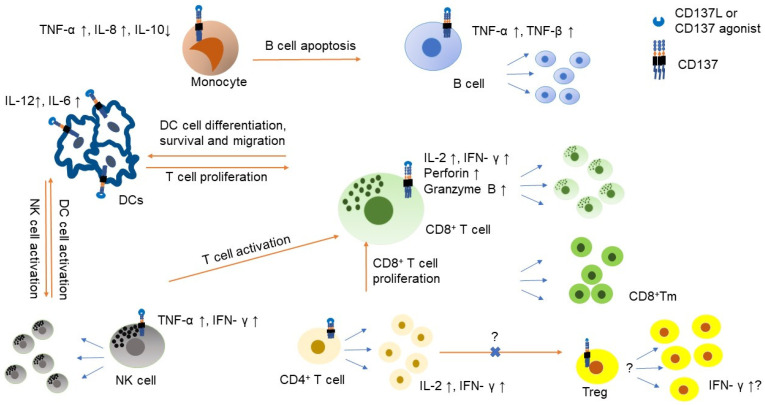
The role of CD137 on immune cells in the tumor microenvironment. CD137 expression is induced by IL-2 or IL-15 on NK cells and stimulation of CD137 on NK cells results in NK cell proliferation and producing IFN-γ, which further support T cell activation [14]. CD137 signaling promotes CD8^+^ T cell proliferation, survival, cytotoxicity and metabolic fitness and can generate CD8^+^ memory T (Tm) cells [10,15]. Both CD4^+^ and CD8^+^ T cells produce IL-2 and IFN-γ upon CD137 stimulation. CD137 function in T regulatory cell (Treg) is controversial and needs to be cautiously interpreted. CD137 stimulation can lead to Treg proliferation but can alter Treg to have cytotoxic or helper effect [16,17]. Transformation of CD4^+^FOXP3^−^ cells to CD4^+^FOXP^+^ can be inhibited by CD137 ligand (CD137L) stimulation [18]. Monocyte also expresses CD137 and stimulation of CD137 induces TNF-α and IL-8 but suppresses IL-10. CD137 stimulation also helps monocyte to differentiate to dendritic cell (DC). It also helps DC to produce IL-12 and IL-6 [19]. However, CD137 stimulation also polarizes monocytes to M2 macrophage and promotes B cell apoptosis. CD137 stimulation to B cell helps B cell to generate TNF-α and TNF-βand it also supports B cell proliferation and survival but does not help class switch [20].

**Table 1 cancers-13-02288-t001:** List of TNFRSF co-stimulatory targeting agents that have been assessed in patients.

Target	Name of IMP	Reference
CD27	Varlilumab	[31]
CD27	MK-5890	[32]
OX40	PF-04518600	[33]
OX40	MEDI0562	[34]
OX40	MEDI6469	[35]
OX40	GSK3174998	[36]
OX40	MOXR0916	[37]
OX40	INCAGN01949	[38]
OX40	BMS-986178	[39]
GITR	TRX518	[40]
GITR	MK-4166	[41]
GITR	MK-1248	[42]
GITR	BMS-986156	[43]
GITR	MEDI1873	[44]
GITR	AMG288	[45]
CD137	Urelumab (BMS-663513)	[46,47]
CD137	Utomilumab (PF-05082566)	[48]

**Table 2 cancers-13-02288-t002:** List of CD137 targeting bispecific molecules in clinical trials.

Company	Target	Format	Development Stage	Reference
PRS343	HER2 × CD137	Anticalin binding protein fused to whole IgG	Phase I	NCT03650348
GEN1046/BNT-311	PD-L1 × CD137	Fc silenced bispecific antibody	Phase I	NCT03917381
RO7122290	FAP × CD137L	Trimeric CD137L and Fab		EUDRACT 2017-003961-83
RO7227166	CD19 × CD137L	Trimeric CD137L and Fab	Phase I	NCT04077723
CB307	PSMA × CD137 × HSA	VH only	Phase I	NCT04839991
PRS-344	PD-L1 × CD137	Anticalin binding protein fused to whole IgG	Phase I	NCT03330561
MCLA-145	PD-L1 × CD137	Full length IgG with two different heavy chains	Phase I	NCT03922204
ES101/INBRX-105	PD-L1 × CD137	Single domain antibody with disabled Fc function	Phase I	NCT04009460
NM21-1480	PD-L1 × CD137 × HSA	3 Fvs linked by linkers	Phase I	NCT04442126
ABL503	PD-L1 × CD137	Anti-PDL1 mAb Fc silenced IgG fused with scFv of anti-CD137	Phase I	NCT04762641
FS222	PD-L1 × CD137	Tetravalent IgG with decreased FcγR binding	Phase I	NCT04740424
FS120	OX40 × CD137	Tetravalent IgG with decreased FcγR binding	Phase I	NCT04648202
GEN1042	CD40 × CD137	Fc silenced bispecific antibody	Phase I/IIa	NCT04083599

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
