# Peer review of "CD137 as an Attractive T Cell Co-Stimulatory Target in the TNFRSF for Immuno-Oncology Drug Development"

_cancers, 2021, doi:10.3390/cancers13102288_

Round 1
Reviewer 1 Report
Remarks to the Author:
There are not many comprehensive reviews available about the CD137 and its important role in immuno-oncology drug discovery. In that aspect, this review article by Kenji Hashimoto is well-written overall. However, since the aim of the CD137 mAb therapy is to stimulate the activity of pre-existing tumor-specific T-cells within the TME, It would be more interesting for readers if the author can write some sentences and cite appropriate references about the expression levels of CD137 on such tumor-infiltrating T cells and the treatment efficacy.
Author Response
Thank you for this comment. To the best knowledge, there is no data suggesting the response of CD137 mAb in association with CD137 expression level in the tumour microenvironment. However, PRS-343 (CD137 x HER2 bispecific) showed some responses in CD8 low tumours. Therefore, following sentence has been added on page 11, line 444.
Indeed, there is no data reporting response of mAb to CD137 in association with CD137 expression level in the tumour microenvironment while some responses were seen in the tumor with low CD8+ T cells for patients treated with PRS343 68.
Reviewer 2 Report
The review article describes a target for drug development, CD137, with a focus on immunotherapy and cancer therapy.
The review is extensive, structured, well-written, uses recent literature for the references. There are two very small typos that I can suggest fixing:
Line 223. Space is missing before "utomilumab".
Line 251. "promotes" is likely "promote" in this context.
Author Response
Thank you for your thorough review and pointing these out. I have added a space before “utomilumab”. I have also removed “s” from promotes.
Reviewer 3 Report
The review by Hashimoto provides a short overview on the CD137/CD137L biology, and focuses then on clinical application of CD137 agonists for cancer immunotherapy. This update on the clinical application of CD137 agonists is very informative.
The title is “CD137 as a target for immuno-oncology drug development” does not reflect the large sections of the review that are on other TNFR family members.
Why does abstract talk about Urelumab but does not mention Utomilumab?
Line 40: “Trimeric ligand binding recruits three receptor monomers which upon ligand binding likewise trimerize to activate intracellular signaling” is not correct. CD137, like all TNFR family members, exists as a trimer, and binding by ligand multimerizes the pre-existing trimers.
Figure 1 is a bit confusing. Is it possible to make the layout a bit more clear. Also, the writing in the Figure is of low resolution. Can reference numbers be added to the texts in the Figure?
The writing in the graphical abstract is of low resolution. “CD8+ T cell” is not entirely readable.
Line 236 “induces IFN-??? and TNF-α generation”
Line 249: Reference 56 is not on CD137 signalling but on the effects of CD137 ligand engagement.
Lines 259, 260: The function of CD137 on Treg is described in a misleading manner. The current state of knowledge is far too complex and too controversial to be expressed in one sentence.
Reference 86 is a retracted article as several other publications by this group.
Minor points:
Line 450: “safety” instead od “safely”.
There are grammatical errors e.g. in the legend to Figure 1: “CD137 stimulation also helps monocyte to differentiate dendric cell (DC) differentiation ….”.
Further, line 269 “only where TAA presents” should be “only where TAA are present”. These are just a few examples. The manuscript should be thoroughly proofread.
The ‘+’ in “CD4+ and CD8+ T cells” should be superscript.
Author Response
Major comments:
1. The title is “CD137 as a target for immuno-oncology drug development” does not reflect the large sections of the review that are on other TNFR family members.
Thank you for this suggestion. The title is now slightly broaden to include T cell co-stimulatory receptors in TNFRSF as “CD137 as an attractive T cell co-stimulatory target in the TNFRSF for immuno-oncology drug development”.
2. Why does abstract talk about Urelumab but does not mention Utomilumab?
Thank you. The following sentence has been added in abstract:
Utomilumab is another CD137 monoclonal antibody to CD137 but is not as potent as urelumab.
3. Line 40: “Trimeric ligand binding recruits three receptor monomers which upon ligand binding likewise trimerize to activate intracellular signaling” is not correct. CD137, like all TNFR family members, exists as a trimer, and binding by ligand multimerizes the pre-existing trimers.
Thank you. The sentence has been amended accordingly as follows:
Like other TNFRSF members, three monomeric CD137s bind to a trimeric CD137L to activate intracellular signalling.
4. Figure 1 is a bit confusing. Is it possible to make the layout a bit more clear. Also, the writing in the Figure is of low resolution. Can reference numbers be added to the texts in the Figure?
The layout of this figure has been amended to be more reader friendly. The figure caption has been amended to explain the figure clearer and citations have been added in the caption. The font size has been changed as well.
5. The writing in the graphical abstract is of low resolution. “CD8+ T cell” is not entirely readable.
The layout has been changed to read sentences clearly.
6. Line 236 “induces IFN-??? and TNF-α generation”
Thank you. It has been revised as IFN- γ
7. Line 249: Reference 56 is not on CD137 signalling but on the effects of CD137 ligand engagement.
Thank you, agree this reference is not appropriate and thus deleted.
8. Lines 259, 260: The function of CD137 on Treg is described in a misleading manner. The current state of knowledge is far too complex and too controversial to be expressed in one sentence.
The following sentences have been added to draw readers attention to interpret Treg function cautiously.
The role of CD137 in Treg cells still needs more investigation particularly in the human system. A few reports suggest that CD137 mAb agonists appear to increase naïve Treg but also converts the function of Treg to cytotoxic or helper T cell aspects and induce a production of IFN-γ (Figure1)16,17. Moreover, CD137L stimulation inhibits antigen- and TGF-β-driven conversion of naïve CD4+FoxP3− T cells into induced Treg cells via stimulation of IFN-γ production by CD4+FoxP3− T cells18. However, high frequency of CD137+ Treg is associated with poor prognosis in lung adenocarcinoma68 and another report suggests that agnostic antibody to CD137 in gastric cancer has little effect on Tregs69. Given that CD137+ Tregs are more immunosuppressive than CD137- Tregs in the non-tumour environment70,71 comprehensive understanding of the role of CD137+ Treg in the tumor microenvironment is warranted.
9. Reference 86 is a retracted article as several other publications by this group.
Thank you. The sentences citing this article have been deleted accordingly.
Minor points:
Line 450: “safety” instead od “safely”.
There are grammatical errors e.g. in the legend to Figure 1: “CD137 stimulation also helps monocyte to differentiate dendric cell (DC) differentiation ….”.
Further, line 269 “only where TAA presents” should be “only where TAA are present”. These are just a few examples. The manuscript should be thoroughly proofread.
The ‘+’ in “CD4+ and CD8+ T cells” should be superscript.
Reply to these minor comments:
Thank you for pointing out these grammatical errors. Once again, proof-reading has been provided and all the above errors are corrected.